# Clinical Outcomes of Single Versus Double Plating in Distal-Third Humeral Fractures Caused by Arm Wrestling: A Retrospective Analysis

**DOI:** 10.3390/medicina58111654

**Published:** 2022-11-15

**Authors:** Jui-Ting Mao, Hao-Wei Chang, Tsung-Li Lin, I-Hao Lin, Chia-Yu Lin, Chin-Jung Hsu

**Affiliations:** 1Department of Orthopaedics, China Medical University Hospital, China Medical University, Taichung 40447, Taiwan; 2Department of Sports Medicine, College of Health Care, China Medical University, Taichung 40447, Taiwan; 3Graduate Institute of Biomedical Sciences, China Medical University, Taichung 40447, Taiwan; 4Spine Center, China Medical University Hospital, China Medical University, Taichung 40447, Taiwan; 5School of Chinese Medicine, China Medical University, Taichung 40447, Taiwan

**Keywords:** single plate, double plate, humerus, fracture, arm wrestling

## Abstract

*Background and Objectives:* Arm wrestling is a simple and popular activity among young people that causes distal-third humeral fractures. However, injury to the young population may cause economic loss; therefore, they need to return to work as soon as possible. Accordingly, we aimed to compare radiological and functional outcomes of distal-third humeral fractures caused by arm wrestling treated with double and single plating. *Materials and Methods:* Thirty-four patients with distal-third humeral fractures caused by arm wrestling were treated between January 2015 and January 2021. They were separated into double- and single-plating groups and treated using a triceps-sparing approach. Regular follow-up was performed to evaluate elbow functionality, range of motion, bone union, and complications; the American Shoulder and Elbow Surgeons score was used for functional assessment. *Results:* Patients treated with single plating exhibited union rate, union time, and elbow range of motion similar to those of patients treated with double plating; however, they exhibited better pain and functional outcomes (American Shoulder and Elbow Surgeons score) at 2 weeks, 1 month, and 3 months postoperatively (84.50 ± 5.01 vs. 61.70 ± 12.53 at 2 weeks, 96.20 ± 2.63 vs. 84.25 ± 14.56 at 1 month, and 100.00 vs. 94.76 ± 9.71 at 3 months, *p* < 0.05). The two groups exhibited no significant differences after 1 year (100.00 vs. 98.54 ± 3.99, *p* < 0.13). The overall complication rate was significantly higher in patients treated with double plating than in those treated with single plating (18.75% vs. 5.56%). Radial nerve palsy was observed in patients in both groups. *Conclusions:* In patients with distal-third humeral fractures caused by arm wrestling, single plating provides a union rate and elbow range of motion similar to those of double plating, with significantly fewer complications and lower surgical time and blood loss with improved early functional outcomes.

## 1. Introduction

The incidence of humeral fractures is 60%, 25.1%, and 10.7% in the middle, proximal, and distal third of the humeral diaphysis, respectively [1]. Arm wrestling is a cause of distal-third humeral fractures. It is a simple and popular activity among young people [2]. During the competition, torsional and axial forces in the humeral shaft may cause significant torque. When the defensive wrestler cannot resist the force, soft tissue damage may occur in the shoulder, elbow, or wrist, as well as fractures [3]. The most common fracture pattern observed after arm wrestling is the extra-articular spiral fracture of the distal third of the humerus [4]. Although the humeral shaft can efficiently tolerate coronal or sagittal plane angulation, non-operative treatment of distal humeral shaft fractures requires more union time, particularly when comparing arm wrestling fractures to other injury mechanisms [5]. Sirbu et al. showed that single plating is an effective treatment option for fractures caused by arm wrestling, with good union rates and functional outcomes [6]. However, some authors have reported superior biomechanical stability with double plating [7,8]. Therefore, the standard management of distal-third humeral fracture remains controversial. We hypothesised that double plating provides more stability in torsional and axial force-induced injuries and has a better functional outcome.

This study aimed to use our data to compare radiological and functional outcomes of distal-third humeral fractures caused by arm wrestling, which were treated with double or single plating.

## 2. Materials and Methods

### 2.1. Patient Population

Overall, 268 patients who experienced distal-third humeral fractures were treated at a level-1 trauma centre from January 2015 to January 2021. The inclusion criteria for the study were fractures that resulted from arm wrestling and those treated using open reduction and internal fixation with double or single plating by two experienced surgeons. The decision to use single or double plating was based on the surgeon’s preference. The exclusion criteria were age <18 years, fractures involving the articular surface or humeral condyle, presence of multiple fracture sites, follow-up of <1 year, cancer history, ligament injury, and preoperative radial palsy. In total, we included 34 patients and categorised them into two groups as follows: single (Group S) and double plating (Group D) (Figure 1).

### 2.2. Surgical Technique

We used a triceps-sparing approach in both groups. First, patients were placed in the decubitus position. Next, a straight skin incision was made, beginning at the centre of the middle to the distal third of the humeral shaft and ending over the ulnar diaphysis. Subsequently, the incision was curved to avoid crossing over the tip of the olecranon. No drainage placements were observed, and we did not perform ulnar nerve transposition in both groups.

### 2.3. Double Plating

We initiated double plating from the ulnar window and subsequently switched to the radial window. In the ulnar window, the ulnar nerve was identified and superficially released through the cubital tunnel while preserving the perineural vessels. In contrast, the triceps fascia was split and mobilised from the lateral intermuscular septum and humerus towards the ulnar side in the radial window. The anconeus was elevated from the posterolateral distal humerus to allow direct visualisation of the fracture.

After the two windows were well-prepared, the fracture site was reduced and temporarily fixed. The metaphyseal plate was first applied on the radial side, followed by the application of the reconstruction plate on the ulnar side (Figure 2). Therefore, the ulnar nerve needed to be tension-free before wound closure.

### 2.4. Single Plating

In this group, only the radial window was exposed with the method described above. In brief, the fracture site was reduced and temporarily fixed. Subsequently, the metaphyseal plate was applied through the radial window (Figure 3).

### 2.5. Postoperative Protocol

We carefully monitored the patients for postoperative distal circulation and neurological conditions. The wound dressings were regularly changed to avoid infections, and stitches were removed at 2 weeks postoperatively. The patients were followed up once a month to assess radiological union and complications (including wound infection, neuroplexia, and painful hardware, among others) until 1 year. Bone union was defined as the presence of three cortex unions on orthogonal radiographic images. At 3 weeks postoperatively, a sling was used for protection during the passive shoulder range of motion assessment, including flexion to 90° and external rotation to 30°, as tolerated. Active shoulder assistive range of motion and active elbow range of motion began at 3–6 weeks postoperatively. The active range of motion of the shoulder and elbow was continued at 6–12 weeks postoperatively when full weight bearing on the injured extremity was also allowed. Finally, we restricted patients from engaging in aggressive exercises until the radiographic union was achieved.

### 2.6. Variables, Data Sources, Measurement, Bias, and Study Size

The data obtained from medical records included basic demographics (age, sex, and underlying disease), fracture type (AO classification), blood loss, surgical duration, and complication rate. American Shoulder and Elbow Surgeons (ASES) score questionnaires were completed by every patient at 2 weeks, 1 month, 3 months, and 1 year postoperatively to assess the clinical and functional outcomes. Elbow flexion and extension range of motion were recorded at 3 and 6 months postoperatively.

### 2.7. Statistical Analysis

Baseline demographics were compared between the two groups using Fisher’s exact test, and the treatment effects were analysed using the Mann–Whitney U test. The improvement rate was defined as the change over time, divided by the baseline values of each variable. All tests were two-tailed, and statistical significance was set at *p* < 0.05. Data were analysed using the Statistical Package for Social Sciences (SPSS) version 22 (SPSS Inc., Chicago, IL, USA).

## 3. Results

Thirty-four patients without underlying disease and preoperative radial nerve palsy who underwent surgery between January 2015 and December 2021 were enrolled. Among them, 18 and 16 patients were in group S and group D, respectively (Table 1). The mean follow-up period was 52 weeks (49–55 weeks). The fracture types included types 12-A, 12-B, and 12-C based on the AO/OTA classification. No significant differences were observed in the variables between the two groups.

Group S had lesser estimated blood loss (205.56 ± 95.32 vs. 293.75 ± 125 mL, *p* = 0.026) and operative time (155.56 ± 36.40 vs. 196.13 ± 56.70 min, *p* = 0.017), which was found to be significant (*p* < 0.05) (Table 2). However, the hospital stay was not significantly different (4.78 ± 0.65 vs. 4.38 ± 0.89 days, *p* = 0.137).

Both group S (Figure 2) and group D (Figure 3) achieved 100% union without significant differences in the union time (90 ± 18.79 vs. 95 ± 16.33 days, *p* = 0.416) (Table 3).

The elbow range of motion was recorded at 3 and 6 months postoperatively. The mean elbow range of motion in flexion and extension was 118.33° ± 13.28° and 8.89° ± 12.31°, respectively, in group S at 3 months postoperatively. However, group D achieved 116.88° ± 17.31° and 6.25° ± 5.63° in flexion and extension, respectively. Furthermore, at 6 months postoperatively, the mean elbow range of motion was 136.67° ± 10.85° and 3.89° ± 4.71° in flexion and extension, respectively, in group S, whereas group D had 133.75° ± 8.47° and 2.5° ± 2.58° in flexion and extension, respectively. The elbow range of motion did not significantly differ in flexion or extension at 3 or 6 months postoperatively (Table 3).

The ASES score was higher (Figure 4), and the visual analogue scale score was lower (Figure 5) in group S, especially at 2 weeks, 1 month, and 3 months postoperatively. Group S had better pain scale and functional outcomes at 2 weeks, 1 month, and 3 months postoperatively than group D (84.50 ± 5.01 vs. 61.70 ± 12.53 at 2 weeks, 96.20 ± 2.63 vs. 84.25 ± 14.56 at 1 month, and 100.00 vs. 94.76 ± 9.71 at 3 months, *p* < 0.05). The two groups did not significantly differ at 1 year postoperatively (100.00 vs. 98.54 ± 3.99, *p* < 0.13). Group S had better overall short-term outcomes than group D but showed no difference in long-term outcomes (Table 3).

The overall complication rate was significantly higher in group D than in group S (*p* < 0.05). Three complications were noted in group D, of which two were due to painful hardware; the remaining patient had radial nerve palsy and recovered after 6 months. However, only one patient in group S developed radial nerve palsy and recovered after 8 months. Furthermore, no patients complained of implant irritation in group S.

## 4. Discussion

Here, in contrast with our hypothesis, single plating provided better short-term functional outcomes and similar union rates in distal-third humeral fractures caused by arm wrestling than double plating. The mechanism of humeral fracture caused by arm wrestling was first described by Brismar in 1975 [9]. Rotational failure due to the internal rotation countering the external rotation was the main cause of distal-third humeral fractures [5]. Ogawa et al. analysed 30 such cases, which were all spiral fractures. Of these, 83%, 16%, and 1% were located in the humeral lower third, middle third, and upper third, respectively [10]. Darren et al. similarly found that the most common injuries were spiral fractures of the distal third of the humerus, which is consistent with the results of our study [4]. Several studies compared the operative and non-operative treatments for humeral shaft fractures and found no differences in union time, ultimate range of motion, and complication rates [11,12]. Bumbaširević et al. presented six cases of humerus fractures caused by arm wrestling and demonstrated that both conservative and surgical approaches were successful treatment methods [3]. However, since most patients with this injury are young adults with high levels of daily activity, who want to return to work as soon as possible, most of them agreed to undergo surgery. To the best of our knowledge, only a few studies have compared surgical single and double plating for this type of injury. In other types of injury, which caused a distal humeral fracture, some studies have found that single plating is sufficient to effectively achieve bone union in extra-articular distal humeral fractures [13,14,15,16,17,18,19,20,21]. However, others showed that double plating provides rigid construction and optimal fracture union, allowing elderly patients to benefit from the early range of motion in intraarticular distal humeral fractures [22,23,24,25]. Prasarn et al. presented 15 cases of extra-articular distal humeral fractures treated with double plating, which achieved an optimal union rate and allowed early range of motion [26]. Consequently, it remains controversial whether single or double plating is the more appropriate method (Table 4).

Comparatively, the perioperative condition was better in patients with single plating than in those with double plating. Patients who were treated with double plating underwent a longer duration of surgery and had higher amounts of estimated blood loss than those treated with single plating. We assume this may be related to greater tissue exposure during the operation. Although the size of the surgical wound was similar in both groups, the ulnar window was exposed only in group D. In this window, it is important to carefully identify the ulnar nerve without causing damage. The fixation of the reconstruction plate also required additional time. After implant fixation, we ensured that the ulnar nerve was tension-free. However, despite the precaution we took with these two aspects, two patients still reported painful hardware. Only one patient developed complications in group S, and this complication was relatively severe. The patient developed radial nerve palsy and fortunately recovered 8 months later. We had assumed that better stability might be necessary when using only a single plate. Therefore, a longer metaphyseal plate was applied on the radial side. The radial nerve is located at the distal third of the humeral shaft, which may be damaged by the longer plate. Although the plate was carefully placed, palsy still occurred during traction. Meloy et al. conducted a retrospective study and showed a 31.25% complication rate in patients treated with double plating and 4.44% in those treated with single plating [13].

Both groups achieved comparable radiological outcomes with a 100% union rate within an average of 3 months. Patients in both groups had a comparable range of motion in the series follow-up. However, group S needed less time to recover and return to their daily activities. The functional score (ASES score) was strongly associated with the pain scale and daily activity. Therefore, we believe that the two groups were predominantly young men with good bone density. Furthermore, rigid fixation can be achieved with a single locking plate. The additional plate plays a minimal role in stability, and its application requires more dissection and exposure. Although we performed this procedure as minimally invasive as possible, dissection in group D resulted in soft tissue damage, which might have caused postoperative adhesions and pain. These may have affected the functional outcomes of these patients. Moreover, after starting a rehabilitation programme, patients showed steady improvement in functional outcomes.

The double-column plating technique is notorious for its high incidence of implant-related complications, such as painful hardware, ulnar neuritis, elbow stiffness, and iatrogenic radial nerve palsy [13]. Here, group D had more instances of painful hardware. We believe that less plating and limited surgical exposure would decrease postoperative implant irritation and soft tissue adhesion. Therefore, we selected a triceps-sparing approach based on the findings of Emmanuel et al., who reported a better elbow range of motion and triceps strength with this approach than those associated with a triceps-splitting approach [27]. Regarding plate configuration in treating distal humeral fractures, a systematic review and meta-analysis conducted by Yu et al. revealed that both the orthogonal and parallel plating methods could achieve successful outcomes with a similarly low number of complications [28]. Therefore, in our study, all patients underwent orthogonal plating.

Our study supports the use of single plating over double plating because the former sufficiently achieves rigid fixation, yields better early functional outcomes, and is associated with a shorter surgical duration, lower blood loss, and fewer complications.

Despite our study’s comparative nature, some limitations need to be highlighted. First, the study was retrospective, and the surgeon selected the fixation strategies for each patient according to the different fracture patterns. Second, the study had a limited sample size.

## 5. Conclusions

In our study, single plating was found to provide similar union rates and elbow range of motion to double plating, with significantly lower surgical times and blood loss with improved early functional outcomes in patients with distal-third humeral fractures caused by arm wrestling.

## Figures and Tables

**Figure 1 medicina-58-01654-f001:**
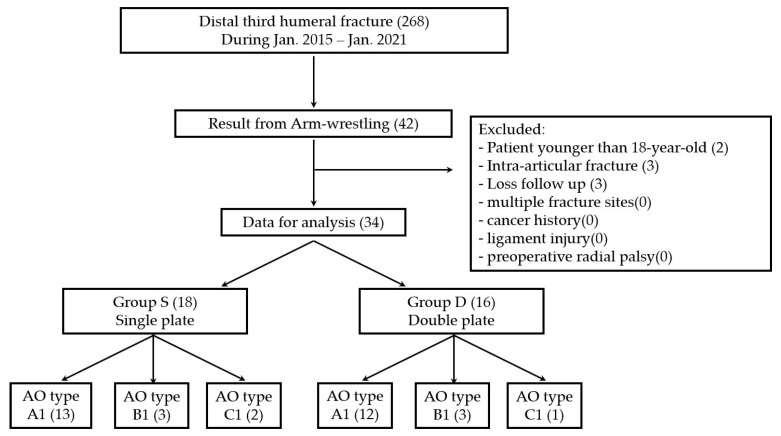
Participants flow. AO = Arbeitsgemeinschaftfür Osteosynthesefragen.

**Figure 2 medicina-58-01654-f002:**
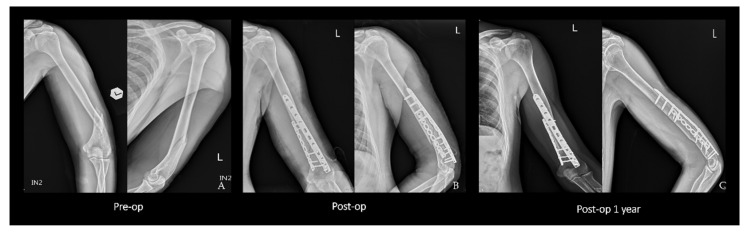
(**A**) Anteroposterior and lateral views showing a spiral fracture. (**B**) Postoperative radiographs showing the fixation with an AO LCP Metaphyseal Plate and AO Reconstruction LCP Plate using a triceps-sparing approach. (**C**) One-year postoperative radiographs revealing bone union. Abbreviations: LCP = locking compression plate. L = left.

**Figure 3 medicina-58-01654-f003:**
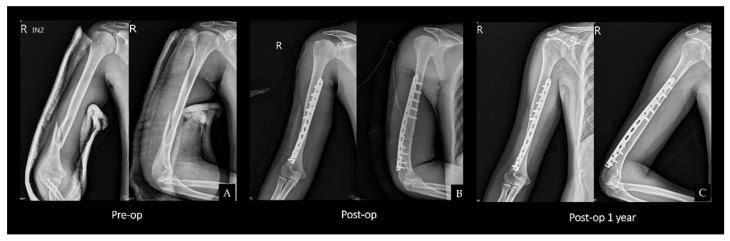
(**A**) Anteroposterior and lateral views showing a spiral fracture. (**B**) Postoperative radiographs showing the fixation with an AO LCP Metaphyseal Plate using a triceps-sparing approach. (**C**) One-year postoperative radiographs revealing bone union. Abbreviations: LCP = locking compression plate. R = right.

**Figure 4 medicina-58-01654-f004:**
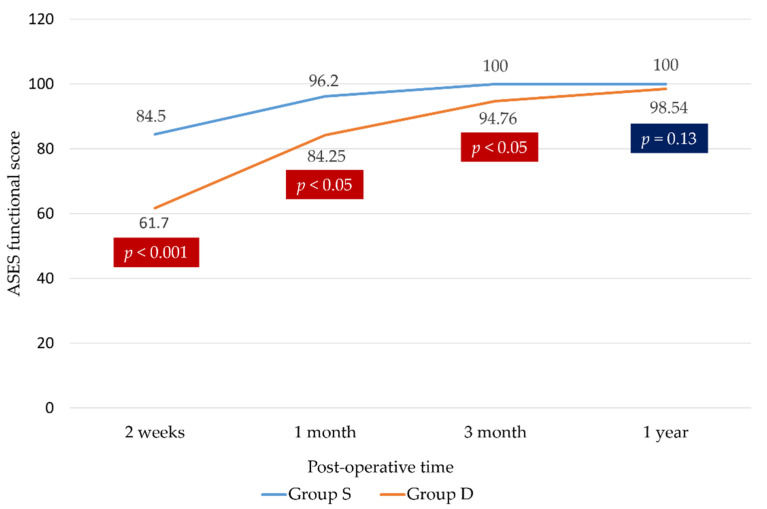
American Shoulder and Elbow Surgeons (ASES) score.

**Figure 5 medicina-58-01654-f005:**
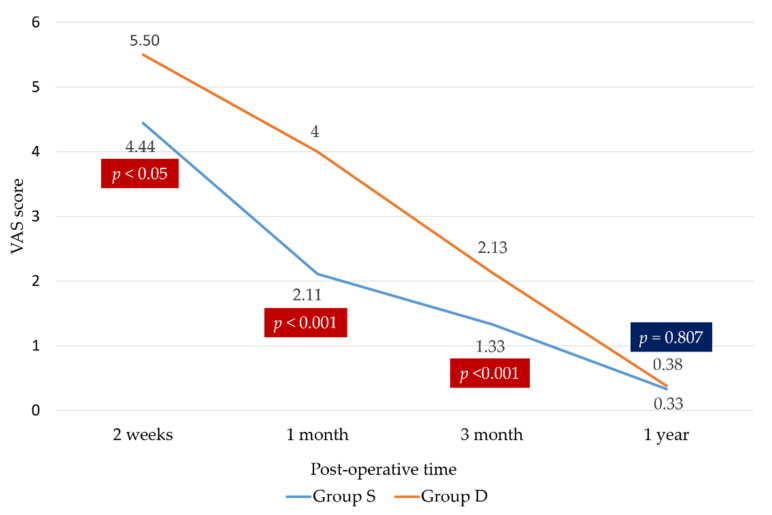
Visual analogue scale (VAS).

**Table 1 medicina-58-01654-t001:** Patients’ demographic data.

	Group S	Group D	*p*-Value
Numbers of patients	18	16	
Mean age (years)	27.56 ± 6.04	28.88 ± 6.12	0.532
Sex			0.732
Male	15 (83.33%)	14 (87.5%)	
Female	3 (16.67%)	2 (12.5%)	
BMI (kg/m^2^)	24.78 ± 1.52	24.00 ± 1.27	0.117
Side			0.800
Right	12	10	
Left	6	6	
Fracture type (AO/OTA classification)			0.880
AO12A1	13	12	
AO12B1	3	3	
AO12C1	2	1	
Underlying disease	0	0	

Abbreviations: BMI = body mass index.

**Table 2 medicina-58-01654-t002:** Perioperative characteristics.

	Group S	Group D	*p*-Value
Operative time (min)	155.56 ± 36.40	196.13 ± 56.70	0.017
Blood loss (mL)	205.56 ± 95.32	293.75 ± 125	0.026
Blood transfusion	0	0	
Complication	1 (5.56%)	3 (18.75%)	<0.05
Hospital stay (days)	4.78 ± 0.65	4.38 ± 0.89	0.137

**Table 3 medicina-58-01654-t003:** Clinical results.

	Group S	Group D	*p*-Value
Union rate	100%	100%	
Union time (days)	90 ± 18.79	95 ± 16.33	0.416
VAS			
2 weeks	4.44 ± 0.86	5.50 ± 0.73	0.001
1 month	2.11 ± 0.58	4.00 ± 0.52	<0.001
3 months	1.33 ± 0.49	2.13 ± 0.34	<0.001
1 year	0.33 ± 0.49	0.38 ± 0.50	0.807
ROM (3 months)			
Flexion (°)	118.33 ± 13.28	116.88 ± 17.31	0.783
Extension (°)	8.89 ± 12.31	6.25 ± 5.63	0.437
ROM (6 months)			
Flexion (°)	136.67 ± 10.85	133.75 ± 8.47	0.393
Extension (°)	3.89 ± 4.71	2.5 ± 2.58	0.303
ASES			
2 weeks	84.50 ± 5.01	61.70 ± 12.53	<0.001
1 month	96.20 ± 2.63	84.25 ± 14.56	0.002
3 months	100	94.76 ± 9.71	0.029
1 year	100	98.54 ± 3.99	0.13

Abbreviations: ROM = range of motion; VAS = visual analogue scale; ASES = American Shoulder and Elbow Surgeons.

**Table 4 medicina-58-01654-t004:** Summary of the literature review of distal-third humeral fractures.

Author	Year	Study Type	Management	Case Number	Conclusion
Gupta et al. [16]	2021	Retrospective study	EADHP	100	Complete union within 3 months: 95% Mean flexion: 123 ± 22° Mean extension: 4.031 ± 6.50°
Ali et al. [17]	2018	Prospective study	EADHP	20	Union time: 17.4 weeks Mean flexion: 127 ± 12.07°
Trikha et al. [18]	2017	Retrospective study	EADHP	36	Complete union within 3 months: 94.44% Mean flexion: 122.9 ± 23° Mean extension: 4.03 ± 6.5°
Kharbanda et al. [19]	2016	Retrospective study	EADHP	20	Mean time to union: 12 weeks Mean flexion: 125°
Scolaro et al. [20]	2014	Retrospective study	3.5-mm PL LCP	40	Achieved union: 95% Reoperation rate: 20%
Capo et al. [21]	2014	Retrospective study	EADHP	19	Union time: 7.3 months Mean flexion: 126° Mean extension: 7°
Meloy et al. [13]	2013	Retrospective comparative study	Double-column plating vs. single pre-contoured PL LCP	105	Single plating offers similar union rates and has significantly fewer complications with improved elbow range of motion
Mark L. Prasarn [26]	2011	Retrospective study	EADHP + 3.3/2.7-mm pelvic recon plate	15	Time to union: 11.5 weeks Mean elbow flexion: 4° Mean extension: 131° Reoperation rate: 13.3%
Watson et al. [25]	2014	Biomechanical study	Standard pre-contoured two-plate locked construct vs. single laterally-placed locked plate	NA	A single plate is biomechanically equivalent to two pre-contoured plates
Scolaro et al. [7]	2014	Biomechanical study	9-hole medial and lateral 3.5 mm DH LCP vs. 6-hole PL LCP	NA	Average bending stiffness and torsional stiffness were significantly greater in 6-hole posterolateral plate
Tejwani et al. [8]	2009	Biomechanical study	One LCP vs. Two reconstruction plates	NA	Double plating provides a more rigid fixation

EADPH = extra-articular distal humerus plating; LCP = locking compression plate; PL = posteriolateral; DH = distal humerus; NA = not applicable.

## Data Availability

All the available data have been presented in this study. Details regarding the data supporting the reported results can be requested at the following e-mail address: jeffrey59835983@gmail.com.

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
