# Peer review of "Clinical Outcomes of Single Versus Double Plating in Distal-Third Humeral Fractures Caused by Arm Wrestling: A Retrospective Analysis"

_medicina, 2022, doi:10.3390/medicina58111654_

Round 1
Reviewer 1 Report
First, I would like to express sincere gratitude to get the opportunity to review your manuscript. The effort of the author is appreciated, as the topic is interesting and promising. Congratulation on your results. Congratulations also for choosing a very specific subject, distal-third humeral fractures caused by arm wrestling.
After assessing the manuscript, the following issues raised my concern or represent suggestions that in my opinion could increase the quality of your manuscript.
Abstract “Drop wrist was noted in patients treated with single plating” - radial nerve palsy was reported by you in the results section of the manuscript in both groups.
Please provide the type of performed study, please be clearer with the patient selection criteria. I understand that this is not an RCT, so based on what criteria it was decided to use a single or double plate technique.
Based on your complications paragraph from the results section of the manuscript, no FRI were encountered in your study but in “Postoperative protocol” you mentioned “Stitches were removed at 2 weeks postoperatively if no surgical-site infection was identified.”, as a personal suggestion just mention that stitches were removed at 2 weeks after surgery.
Reviewer 2 Report
Title:
Ok
Abstract:
Background and Objectives: We aimed to compare radiological and functional outcomes of distal-third humeral fractures caused by arm wrestling treated with double plating and single plating.
This sentence is the aim of the study, no background is provided here. Consider adding 1-2 sentences that will emphasize the importance or rationale for this study.
Why arm wrestling? Why this mechanism is important or different from other causes?
Line 24: they exhibited better pain and functional outcomes at 2 weeks, 1 month, and 3 months postoperatively.
provide information regarding functional outcomes (i.e., ASES score)
Line 26: The overall complication rate was significantly higher in patients treated with double plating. Percentage?
Introduction:
Line 37-39: It is a simple and popular activity among young people, and it usually consists of two persons sitting across each other at a table, placing their elbows on it and joining hands…
These sentences can be removed and in fact the introduction can start from line 43.
At the end of the introduction please add hypothesis.
Methods:
Regarding inclusion\exclusion criteria, what about concomitant soft tissue injuries? Pathological fractures, please elaborate and clarify the inclusion\exclusion criteria.
Please add information regarding the decision to use single or double plating, as this study is retrospective from 1 medical center, why there were 2 different approaches to treat these fractures?
Please clarify.
Add information regarding presence of preoperative radial nerve palsy in the study cohort.
Surgical technique:
Add information regarding plate size considerations (length, thickness)
Why did you used AO LCP Metaphyseal Plate and AO Reconstruction Plate and not 2 LCP plates?
Did you use a drain? Please clarify.
Did you perform ulnar nerve transposition? Please clarify.
Line 110: What complications? Please specify
Results:
Line 123-126: the authors can refer to the demographic table and shorten the text.
Table 1 is missing underlying disease as mentioned in line 110.
Line 141-147- the authors can refer to the demographic table and shorte”n the text.
In the abstract the authors stated: “Drop wrist was noted in patients treated with single plating” However, in line 167: “Only one patient in group S developed radial nerve palsy and recovered after 8 months” Please clarify.
Additionally, line 164: “The overall complication rate was significantly higher in group D than in group S (p < 0.05).
In the D group you mention three complication vs 1 complication in the S group.
This paragraph Is unclear. Please correct or clarify.
Discussion:
In general, the discussion needs editing, mainly in light of repetitiveness.
The discussion should include the following:
1.Summarize the key findings in clear and concise language
2. Acknowledge when a hypothesis may be incorrect
3. Place your study within the context of previous studies
4. Discuss potential future research
5. Provide the reader with a “take-away” statement to end the manuscript
Line 178-180: “Several studies compared the operative and non-operative treatments for humeral shaft fractures and found no differences in union time, ultimate range of motion, and complication rates…”
These studies addressed humeral fractures following arm wrestling? This is not clear. Please clarify.
Table 4 : These studies addressed humeral fractures following arm wrestling? This is not clear. Please clarify.
If not, please see my previous comment regarding the importance of this mechanism of injury.
I'm not sure how relevant the table is to the article, it can be omitted and only relevant studies that compared a single plate with two plates can be discussed in the text.
Line 196-210 , please address previous studies and reported comlication rate and compare to the current study.
Line 215-221: this paragraph is not clearly supported with the study results. Please omit or edit.
Line 235 : please see my previous comment regarding complications’ rate.
Line 239-241: please remove.
Line 243: “Arm wrestling is a popular activity which can cause serious complications, especially 243 distal humeral fractures” remove from the conclusion. State only the main finding and “take home message” that can be concluded from the study’s’ results.
Round 2
Reviewer 1 Report
Dear authors,
The performed changes definitely improved the quality of the manuscript.
Author Response
Point 1: The performed changes definitely improved the quality of the manuscript.
Response 1: Thank you for your comment regarding our manuscript.

Reviewer 2 Report
Abstract:
Line 24: they exhibited better pain and functional outcomes at 2 weeks, 1 month, and 3 months postoperatively.
Please add and provide information regarding functional outcomes (i.e., ASES score) – results – scores +\- STD of both groups.
Methods:
Regarding inclusion\exclusion criteria,
Add information regarding presence of preoperative radial nerve palsy in the study cohort.
And please explain why this was an exclusion criterion.
Author Response
Point 1: Line 24: they exhibited better pain and functional outcomes at 2 weeks, 1 month, and 3 months postoperatively.
Please add and provide information regarding functional outcomes (i.e., ASES score) – results – scores +\- STD of both groups.
Response 1: Thank you for your valuable suggestion. We have added the information:
“however, they exhibited better pain and functional outcomes (American Shoulder and Elbow Surgeons score) at 2 weeks, 1 month, and 3 months postoperatively (84.50 ± 5.01 vs. 61.70 ± 12.53 at 2 weeks, 96.20 ± 2.63 vs. 84.25 ± 14.56 at 1 month, and 100.00 vs. 94.76 ± 9.71 at 3 months, p<0.05). The two groups exhibited no significant differences after 1 year (100.00 vs 98.54 ± 3.99, p<0.13).” (Line 26-30)
We also added this information in line 164-166.
Point 2: Regarding inclusion\exclusion criteria. Add information regarding presence of preoperative radial nerve palsy in the study cohort. And please explain why this was an exclusion criterion.
Response 2: Thank you for your careful review. We excluded those with preoperative radial nerve palsy in the study cohort in order to avoid evaluations for related postoperative complications. Fortunately, in this study, there were no cases of preoperative radial nerve palsy associated with distal humerus fracture due to arm wrestling.
